# FCG Modelling Considering the Combined Effects of Cyclic Plastic Deformation and Growth of Micro-Voids

**DOI:** 10.3390/ma14154303

**Published:** 2021-07-31

**Authors:** Edmundo R. Sérgio, Fernando V. Antunes, Micael F. Borges, Diogo M. Neto

**Affiliations:** Centre for Mechanical Engineering, Materials and Processes (CEMMPRE), Department of Mechanical Engineering, University of Coimbra, Pinhal de Marrocos, 3030-788 Coimbra, Portugal; edmundo.sergio@uc.pt (E.R.S.); micaelfriasborges@outlook.pt (M.F.B.); diogo.neto@dem.uc.pt (D.M.N.)

**Keywords:** fatigue crack growth, crack tip plastic deformation, GTN damage model, crack closure

## Abstract

Fatigue is one of the most prevalent mechanisms of failure. Thus, the evaluation of the fatigue crack growth process is fundamental in engineering applications subjected to cyclic loads. The fatigue crack growth rate is usually accessed through the *da/dN-*Δ*K* curves, which have some well-known limitations. In this study a numerical model that uses the cyclic plastic strain at the crack tip to predict *da/dN* was coupled with the Gurson–Tvergaard–Needleman (GTN) damage model. The crack propagation process occurs, by node release, when the cumulative plastic strain reaches a critical value. The GTN model is used to account for the material degradation due to the growth of micro-voids process, which affects fatigue crack growth. Predictions with GTN are compared with the ones obtained without this ductile fracture model. Crack closure was studied in order to justify the lower values of *da/dN* obtained in the model with GTN, when compared with the results without GTN, for lower Δ*K* values. Finally, the accuracy of both variants of the numerical model is accessed through the comparison with experimental results.

## 1. Introduction

Design against fatigue is fundamental in components and structures submitted to cyclic loads. The damage tolerance approach involves the designing of structural components with certain allowance for small cracks, whose presence must be checked by periodic inspection. It is particularly recommended for manufacturing industries where defects are unavoidable, such as the case of welding, casting [1] or additive manufacturing [2]. The ability to model and predict fatigue crack growth (FCG) rate precisely is one of the key aspects of damage tolerance approach.

The modelling of FCG must be based on the knowledge of the fundamental damage mechanisms acting at the crack tip. Cyclic plastic deformation is usually assumed to be the main damage mechanism [3,4]. However, since FCG rate spans over seven orders of magnitude, other mechanisms may be expected. Environmental damage is supposed to have a significant contribution near-threshold [5,6]. At higher load levels other mechanisms may be expected, namely the growth and coalescence of micro-voids [7] and brittle failure.

In previous works of the authors, FCG was predicted numerically assuming that cyclic plastic deformation, quantified by cumulative plastic strain, is the crack driving force. This approach was able to predict qualitatively the effects of Δ*K* [8], maximum and minimum loads, stress ratio [9], overloads [10] and complex load patterns [11]. However, the comparison with experimental results showed that the effect of stress ratio was lower than that obtained experimentally, particularly for Ti-6Al-4V alloy [10]. Besides, the slopes of experimental *da/dN*-Δ*K* curves were found to be higher than the slopes predicted numerically for the Ti-6Al-4V alloy [12] and the 2024-T351 aluminium alloy [8]. In other words, an anti-clockwise rotation of predicted *da/dN*-Δ*K* curve (Paris regime) is needed to improve the fitting to experimental results. These difficulties indicated that cyclic plastic deformation does not characterise completely the crack tip damage, and that other mechanisms are needed. Therefore, in this work, the growth and coalescence of micro-voids was also included in the analysis, providing a better modelling of the FCG phenomenon. Figure 1 is a schematic illustration of the approach followed here to study FCG. The independent parameters, which include the material, geometry, load and environment, determine *da/dN*. Cyclic plastic deformation at the crack tip is assumed to be the crack driving force, but the growth and coalescence of micro-voids is expected to affect it. This mechanism greatly depends on stress triaxiality, which is affected by geometrical parameters including stress state and crack length. The crack closure phenomenon, which is linked to residual plastic deformation, affects the effective load range felt at the crack tip.

The growth and coalescence of micro-voids is a mechanism usually associated with ductile failure of metals. However, it can also affect FCG since metallic materials have intrinsic defects which can grow, while new defects can also be nucleated. Under these conditions, FCG can be accelerated. However, studies of FCG with a focus on the growth, nucleation and coalescence of micro-voids are not common in the literature. These mechanisms are rather associated with ductile tearing [13] and micro-crack initiation [14]. The appearance of micro-voids, as well as the processes involving them, influence the behaviour of the material. The quantification of the influence of the micro-voids is performed through an entity called damage [15]. However, damage is not a simple variable as it is directly related with the stress state and plastic strain [16]. Additionally, it is an internal and cumulative entity that cannot be measured directly [17]. Thus, this damage accumulation mechanism is usually modelled with the so called damage models, GTN (Gurson–Needleman–Tvergaard) being one of the most famous [18]. Note that the damage accumulation is not accounted for only by failure criteria [19] but also for the decrease in material stiffness, strength and a reduction in the remaining ductility [20]. 

The main objective of this study is assessing the effect of accounting for the GTN model on the predicted *da/dN*. The FCG rate is assumed to be controlled by cumulative plastic strain at the crack tip, which is affected by the material damage defined through the GTN model. Gurson [21], using micro-mechanical considerations, introduced a yield potential for materials containing micro-voids and, from the study of a single cavity in a elastically perfectly plastic, void-free matrix, derived a void evolution law [22], which is given by: (1)$$\dot{f}=\left(1-f\right){{\dot{\varepsilon }}_{v}}^{p}=\left(f-f^{2}\right)\dot{\gamma }\sigma_{y}\sinh\left(\frac{3p}{2\sigma_{y}}\right)\;$$
where f is the void volume fraction (porosity), ${{\dot{\varepsilon }}_{v}}^{p}$ the volumetric plastic strain rate, $\dot{\gamma }$ is the plastic multiplier, p the hydrostatic pressure and $\sigma_{y}$ the flow stress given by the hardening law. 

Tvergaard [23,24], by introducing the void interaction parameters ($q_{1}$,$\;q_{2}$ and $q_{3}$), modified the Gurson plastic potential to account for the interactions between neighbouring voids, resulting in the following yield potential:(2)$$\phi ={\left(\frac{{\bar{\sigma }}^{2}}{\sigma_{y}}\right)}^{2}+2q_{1}f\cosh\left(q_{2}\frac{\mathrm{tr}\;\sigma }{2\sigma_{y}}\right)-1-q_{3}f^{2}\;\;$$
where $\bar{\sigma }$ is the von Mises equivalent stress, while tr σ denotes the trace of the stress tensor. 

Needleman and Chu [25] introduced the possibility of void nucleation in a statistical fashion, considering a normal distribution around a mean plastic strain,$\;\varepsilon_{N}$, with a standard deviation, $s_{N}$, and a maximum nucleation amplitude, $f_{N}$. Thus, the void evolution is now represented by the sum of the void growth law, given by Equation (1), and the void nucleation rate $f^{n}$: (3)$${\dot{f^{n}}}_{\;}=\frac{f_{N}}{s_{N}\sqrt{2\pi }}\exp\left[-\frac{1}{2}{\left(\frac{{\bar{\varepsilon }}^{n}-\varepsilon_{N}}{s_{N}}\right)}^{2}\;\right]{\dot{\bar{\varepsilon }}}^{\;p}$$

The GTN model is categorised as a micromechanical-based coupled model. This means that the material constitutive equations are affected by the damage accumulation due to micro void operations. Additionally, there is an intrinsic coupling relation between porosity, plastic strain [26] and stress triaxiality [27]. The evolution of the porosity defined according to the GTN model takes into account the stress triaxiality [28]. The high degree of stress triaxiality occurs near the crack tip [29] since the stress state in this central region is essentially plane strain. Near the free surface, the stress triaxiality is lower due to the pure plane stress at the free surface. [30]. The influence of the triaxiality level on the porosity evolution predicted by GTN is analyzed in this study. The stress triaxiality, which ranges between 0 (pure shear) and 5 or 6 (sharp notches), is defined by the ratio between the hydrostatic and von Mises equivalent stresses [31]: (4)$$T=\frac{p}{\sigma_{eq}}=\frac{\frac{\sigma_{xx}+\sigma_{yy}+\sigma_{zz}}{3}}{\sqrt{\frac{1}{2}\left[{\left(\sigma_{xx}-\sigma_{yy}\right)}^{2}-{\left(\sigma_{yy}-\sigma_{zz}\right)}^{2}-{\left(\sigma_{zz}-\sigma_{xx}\right)}^{2}+3\left(\sigma_{xy}^{2}+\sigma_{xz}^{2}+\sigma_{yz}^{2}\right)\right]\;}}$$

Stress triaxiality appears in several other fields of study. Chen et al. [32] studied the variations of stress-triaxiality during the initial and stable phases of ductile crack growth. Wang et al. [33] used this parameter in the evaluation of the stress states near weld chords in the study of ductile failure of tubular joints. Anvari et al. [34] also used this concept on the study of ductile crack growth using rate-sensitive and triaxiality-dependent cohesive elements. 

The article is organised as follows: in section two there is a description of the applied material constitutive model and respective parameters, deformable body dimensions and discretization, loading case and fatigue crack growth algorithm. Section three presents the obtained results, with the entities analysed being chain related. Section four presents a discussion on the obtained *da/dN-*Δ*K* curves with GTN, without GTN and experimentally. Finally, the conclusions reached are enumerated. 

## 2. Numerical Model 

All numerical simulations were performed with the in-house finite element code DD3IMP, originally developed to simulate deep-drawing processes [35,36]. The mechanical behaviour of the specimen was modelled by a temperature-independent elastoplastic constitutive model. The temperature rise due to the heat generated by plastic deformation is neglected in the model. 

### 2.1. Material Constitutive Model

In this study the 2024-T351 aluminium alloy was considered, and a continuous mechanics approach was followed in the numerical study. The isotropic elastic behaviour of the material is described through Hooke’s law. Von Mises yield criterion was used to define the yield surface. The hardening behaviour is described by Swift and Armstrong–Frederick [20] laws, given by Equations (5) and (6), respectively: (5)$$\sigma_{y}\left({\bar{\varepsilon }}^{p}\right)=k{\left({\left(\frac{Y_{0}}{k}\right)}^{\frac{1}{n}}+{\bar{\varepsilon }}^{p}\right)}^{n}$$
(6)$$\dot{X}=C_{X}\left[\frac{X_{\mathrm{Sat}}}{\bar{\sigma }}\left(\sigma^{\prime }-X\right)\right]{\dot{\bar{\varepsilon }}}^{p}\;\mathrm{with}\;\dot{X}\left(0\right)=0,\;$$
where for the Swift law: *Y*_0_, *k*, and *n* are the material parameters and ${\bar{\varepsilon }}^{p}$ is the equivalent plastic strain. In the case of Armstrong–Frederick law: X is the back stress tensor, $X_{\mathrm{Sat}}$ and $C_{X}$ are material parameters, $\sigma^{\prime }$ is the deviatoric component of the Cauchy stress tensor, and ${\dot{\bar{\varepsilon }}}^{p}$ is the equivalent plastic strain rate. The material constitutive parameters are presented in Table 1, which were obtained by minimization of the difference between numerical and experimental data from low cycle fatigue tests [21].

The GTN parameters related with the growth of voids were chosen based on the existing literature regarding this aluminium alloy [37]; these are presented in Table 2. The initial porosity (*f*_0_) was overestimated to the largest value range in order to overcome the inexistence of nucleation.

### 2.2. Geometry, Mesh and Loading Case

The geometry and main dimensions of the deformable body, a compact tension (CT) specimen in accordance with ASTM E647 standard [38], are shown in Figure 2. Due to the existent symmetry on the crack plane, only the upper part of the specimen was modelled. A thickness of 0.1 mm was considered to reduce the computational cost. Only plain strain conditions, which were imposed by constraining out of plane displacements, have been studied. This way, the results are independent of the specimen thickness. Usually, the specimens are submitted to pre-cracking, which causes the crack tip to move away from the notch; therefore, the geometry of the notch does not affect FCG and was not modelled numerically.

The specimen was loaded, considering a single point force applied on the specimen hole, with a constant amplitude cyclic load. Mode I is considered and the variation range was set between *F*_min_ = 4.17 N and *F*_max_ = 41.7 N, resulting in a stress ratio, *R* = 0.1. The deformable body geometry was discretised with 8-nodelinear hexahedral finite elements, while a selective reduced integration technique was adopted to avoid volumetric locking [39]. The region surrounding the crack growth path is meshed with elements of 8 μm, which allows to accurately evaluate the strong gradients of stresses and strains at this zone [40]. To reduce the computation cost, the outer region was coarser meshed. The specimen thickness is defined by only one layer of elements. In the end, 7287 finite elements and 14,918 nodes were used.

### 2.3. FCG Algorithm

The fatigue crack growth process is modelled through a node release strategy [8]. This occurs when the plastic strain, measured at the Gauss points and averaged at the node containing the crack tip, reaches a critical value, $\varepsilon_{c}^{p}$. This parameter is supposed to be a material property, and based on a previous study [8], it was considered that $\varepsilon_{c}^{p}=1.1$. Note that this value corresponds to a plastic strain of 110%. The fatigue crack growth rate is obtained from the ratio between the crack increment (8 μm, which is the element size) and the number of load cycles, Δ*N*, required to reach the critical value of plastic strain. A total plastic strain (TPS) approach was followed, which means that the plastic strain, and porosity, accumulated in the previous load cycles, in a certain node are not reset when propagation occurs.
(7)$$\frac{da}{dN}=\frac{8\;\mu m}{\Delta N}$$

Note that the FCG rate is assumed constant between crack increments. Since the crack propagation rate is usually relatively low (<1 μm/cycle), the numerical analysis of the crack growth is simplified by considering different sizes for the initial straight crack. Initial crack sizes, $a_{0}$, of 5, 9, 11.5, 16.5, 19 and 21.5 mm were considered. Some crack propagation is required to stabilise the cyclic plastic deformation and the crack closure level—only after that the FCG is evaluated. Finally, the contact between the flanks of the crack is modelled considering a rigid plane surface aligned with the crack symmetry plane.

## 3. Results

### 3.1. Fatigue Crack Growth Rate

Figure 3 shows the *da/dN*-Δ*K* curves predicted numerically with and without the GTN model. The horizontal and vertical axes are presented in log-log scales, as usual. The *da/dN*-Δ*K* curve without GTN follows an approximately linear trend in log-log scale, through all Δ*K* values studied, with a Paris law coefficient of *m* = 2.62. The inclusion of GTN produced significant changes in *da/dN*. For low values of Δ*K* there is a decrease in *da/dN* with the inclusion of the growth of micro-voids in the model, while for high values of Δ*K* the opposite trend is observed. The inversion of behaviour occurs at about Δ*K* = 11.5 MPa.m^0.5^ The model with GTN roughly follows a linear trend for lower values of Δ*K*, but the linearity disappears when the full range of Δ*K* is included. The Paris law coefficient is also higher (*m* = 3.36).

The nucleation, growth and coalescence of micro-void phenomena are supposed to deteriorate the material stiffness. Moreover, this ductile damage model is directly related to the plastic deformation, which is important at the crack tip. Thus, it was expected that the introduction of the GTN damage model would result in an increase in the fatigue crack growth (FCG) rate. Nevertheless, the growth of micro-voids in the model may have a protective behaviour, reducing the FCG rate. An explanation for the odd behaviour observed at relatively low values of Δ*K* is required.

### 3.2. Cumulative Plastic Strain

To explain the influence of the GTN model on FCG rate, both plastic strain and crack closure were studied for two different values of stress intensity factor. Accordingly, two initial crack lengths are evaluated, namely *a*_0_ = 11.5 mm, which corresponds to a stage where the model with GTN predicts a lower *da/dN* than the model without GTN; and *a*_0_ = 21.5 mm, which corresponds to the final phase of the crack growth, where the FCG rate is higher with GTN. Figure 4a shows the evolution of the plastic strain through the period between the 25th and 26th crack propagations for both models (with and without GTN). Time was reset on the instant were the previous propagation occurred, so that propagations from both models could be compared. The results show that the 25th crack propagation causes the plastic strain to decay to a local minimum. This entity is evaluated at the node containing the crack tip; thus, when propagation occurs, the crack tip advances to the following node where the plastic strain is still small. The subsequent load cycles cause the plastic strain to increase in a cumulative way. However, the plastic strain clearly grows faster in the model without GTN, and the critical plastic strain is achieved faster. Once the critical strain, $\varepsilon_{c}^{p}$, is achieved, node release occurs for both models, and a new accumulation begins.

Figure 4b presents the plastic strain evolution at the crack tip during a single load cycle, immediately before the 26th crack propagation, comparing the two models. The initial constant value is due to crack closure and consequent absence of plastic deformation at the crack tip. The plastic deformation starts later in the model with GTN, which may be explained by different crack closure levels. The increase in load up to the maximum value produces an accumulated plastic strain, which is higher in the model without GTN. The same trend is followed in the unloading phase. This explains the higher slope of the plastic strain curve observed in Figure 4a for the model without GTN.

Figure 5a,b present analogous results but for *a*_0_ = 21.5 mm, corresponding to the period between the 36th and the 37th crack propagations. Different propagations were chosen because the crack growth stabilization is slower for higher initial crack lengths. The minimum values of plastic strain after the 36th node release are higher than the ones observed in Figure 4a. They denote the plastic strains accumulated while the Gauss point is located ahead of crack tip. Since the size of the cyclic plastic zone increases with Δ*K*, larger initial plastic strains are expected for higher Δ*K* levels. Moreover, the inclusion of GTN also results in a higher cumulative plastic strain in the crack tip at the beginning of the propagations, which is linked to the increase in plastic strain produced by GTN. The application of the load cycles leads to an increase in the plastic strain in the crack tip (see Figure 5a). However, it grows faster using the model with GTN. Regarding the evolution of the plastic strain in the crack tip during a single load cycle, the results in Figure 5b show that plastic strain starts to increase at approximately the same time for both models. However, the increase in the plastic strain is much faster using the GTN model. Thus, the inclusion of the damage model has a detrimental effect on the material strength, increasing the plastic strain rate during the loading. Similar to the previous case, the same trend is verified in the unloading stage.

### 3.3. Size of the Plastic Zone at the Crack Tip

The results shown in Figure 4a and Figure 5a indicate that, at the beginning of a new propagation, the plastic strain is higher in the case of the model with GTN. As a TPS approach is followed, this may be explained by the occurrence of higher plastic zones at the crack tip, which lead to sooner increments of plastic strain in the farthest nodes. To prove this hypothesis the distance between the node containing the crack tip and the first node exhibiting no plastic strain was measured in the propagation direction. The size of the plastic zone is presented in Figure 6, comparing the two crack lengths (*a*_0_ = 11.5 mm and *a*_0_ = 21.5 mm) previously analysed, as well as both models—with and without damage model. The horizontal axis presents the fraction of load cycles required to reach the critical plastic strain. Since the plastic zone size is significatively larger than the crack increment (8 μm), it is approximately constant within each propagation. On the other hand, for both initial crack lengths analysed, the model with GTN leads to larger plastic zone sizes, explaining the higher initial plastic strain at the beginning of a new propagation. Additionally, higher *a*_0_ also leads to higher dimensions of the plastic zone due to the higher Δ*K* levels occurring at the crack tip.

### 3.4. Plasticity Induced Crack Closure

The trends followed by the plastic strain explain the differences in the behaviour of the *da/dN*-Δ*K* curves. However, the behaviour of plastic deformation itself also requires an explanation. Figure 7 presents the crack tip open displacement (CTOD) measured at the first node behind the crack tip, at a distance of 8 μm. Figure 7a shows the CTOD in the last load cycle before the 26th propagation for *a*_0_ = 11.5 mm, while Figure 7b shows analogous results but for the 37th propagation of *a*_0_ = 21.5 mm. The CTOD curves presented in Figure 7 were evaluated for the same load cycles for which the plastic strain evolution was evaluated in Figure 4b and Figure 5b. Considering the damage model, lower CTOD levels are predicted for both initial crack lengths. This can be explained through the fact that the higher plastic strain induced by the GTN results in higher plastic wakes at the crack flanks and consequently, a higher trend to close the crack. The crack closure reduces the effective load range, protecting the material from FCG since the crack only grows when it is open. The lower growth rate of plastic strain for *a*_0_ = 11.5 mm matches the higher closure level attained with the model considering GTN. Note that, without GTN, there is no crack closure. On the other hand, for *a*_0_ = 21.5 mm the crack closure is very small, even with GTN. Thus, as the crack closure ceases to protect the material, the higher plastic strain achieved with the GTN model causes a faster FCG rate.

The crack closure level was evaluated during an entire propagation for both initial crack lengths, with and without GTN. The crack closure level was quantified, over the load increments, considering the contact status of the first node behind crack tip and using the parameter:(8)$$U^{*}=\frac{F_{\mathrm{open}}-F_{\min}}{F_{\max}-F_{\min}}$$
where *F*_open_ is the crack opening load, *F*_min_ is the minimum load and *F*_max_ is the maximum load. This parameter quantifies the fraction of load cycle during which the crack is closed.

Figure 8a presents the crack closure evolution between the 25th and the 26th crack propagations of *a*_0_ = 11.5 mm, comparing the predictions with and without the damage model. A transient behaviour is registered at the beginning, consisting of a fast increase followed by a progressive decrease to a stable value. Initially, crack closure rises due to the accumulation of plastic strain and formation of residual plastic wake. During the transient stage, crack closure is very sensible to the point where it is measured. The successive load cycles cause the crack tip to blunt, reducing the crack closure level. Note that the trend of the crack closure during the loading cycles is the same for both models; there is only a vertical shift of the curve referring to the model considering GTN. However, while the model without GTN completely loses crack closure, the model with GTN stabilises at *U** = 20%. Other authors also found no closure in their numerical studies without GTN, namely Zhao and Tong [41] in a CT specimen and Vor et al. [42] at the centre of a 3D CT specimen.

Figure 8b shows similar results but between the 36th and the 37th node releases of *a*_0_ = 21.5 mm. For this initial crack length, the propagation with GTN takes considerably less cycles. Thus, it was impossible to measure crack closure at the very same point during propagation. During the load cycling, crack closure is higher for the model with GTN. Nevertheless, the trend followed by both models is different from the one registered for *a*_0_ = 11.5 mm. The initial peak is now more pronounced, which is due to the higher plastic strain produced by the harsher stress field at the crack tip induced by higher Δ*K* level. The subsequent decrease of *U** is a blunting effect caused by the cyclic loading, which moves the node behind crack tip [43]. This phenomenon is related with strain ratcheting, and greatly depends on material, being more relevant for material models comprising the kinematic hardening component. It also depends on stress state, being more relevant for plane strain state, as is the case in [43]. The numerical model comprises both conditions; thus, this effect is expected to be relevant, causing the crack closure to eventually cease. Even if the crack closure remains higher for the model with the GTN, the protection to the material is reduced, with it approaching the levels showed by the model without GTN. As the protection decays the higher tendency to accumulate plastic strain, due to the deterioration of the material through porosity, comes on top. Crack closure is, therefore, the key to understanding the *da/dN* behaviour of both models.

Finally, crack closure was disabled in the model with GTN. This is achieved numerically by deactivating the contact of the nodes that cover the crack flanks. Figure 9a presents the plastic strain evolution, throughout the time period between the 25th and 26th propagations, for the two specifications of the model with GTN—with and without contact—for *a*_0_ = 11.5 mm. Figure 9b presents analogous results but for the plastic strain build-up at the single load cycle, immediately before the 26th propagation. Figure 9a shows that the plastic strain starts from similar levels after the 25th node release. The subsequent increase in plastic strain is much faster without crack closure. Thus, the *da/dN* differences are only a consequence of the much faster accumulation of plastic strain. Figure 9b shows that plastic strain starts to rise much sooner without crack closure. In other words, crack closure delays the start of the accumulation of plastic strain at each loading cycle. This means that the contact of the crack flanks reduces the range of effective stress at the crack tip. Note that plastic strain is a nonlinear entity; thus, during the growing stage it follows a nonlinear trend, but this trend is essentially the same for both variations of the model, as indicated by the dashed lines. With crack closure, as its start is delayed, when maximum force is achieved the accumulation is just at a different stage of the same path. Moreover, during the unloading phase the same trend is followed. However, this time crack closure influences the last part of the loading cycle, planning the accumulation of plastic strain.

Figure 10 shows *da/dN-*Δ*K* results for the model considering the GTN model, with and without crack closure, in log-log scales. The models without crack closure produce higher values of *da/dN*, which is in agreement with the result in Figure 9. The dramatic effect of disabling crack closure for *a*_0_ = 11.5 mm is attenuated for *a*_0_ = 21.5 mm. As discussed before, the effect of crack closure is of less importance for *a*_0_ = 21.5 mm. Thus, for higher values of Δ*K*, the FCG rate with and without crack closure would be very close, as shown in Figure 10. Moreover, for these values of Δ*K* the propagation rate is almost independent of crack closure.

### 3.5. Porosity versus Plastic Deformation

The plastic strain arising at the crack tip leads to an accumulation of damage defined in terms of porosity growth. In other words, the plastic strain is the driving force of porosity accumulation. Thus, the implementation of the GTN model, in the existing FCG model, was expected to result in a growth of damage in accordance with the evolution of plastic strain at the crack tip. To verify this relation, both entities were analysed at the crack tip node. Figure 11 shows the evolution of porosity with plastic strain, during all load cycles of a single propagation, for three different values of initial crack length, namely 5, 11.5 and 21.5 mm. Note that the results are presented in natural scales. There is a general trend for the increase of porosity with plastic strain. For *a*_0_ = 5 mm there is an initial nonlinear increase in porosity, followed by a saturation zone. This means that the plastic strain increases but the porosity does not increase. In the case of *a*_0_ = 11.5 mm, the initial nonlinear increase is followed by a linear zone where the increase in porosity is proportional to the increase of plastic strain. For *a*_0_ = 21.5 mm there is neither initial transient regime nor saturation. The maximum porosity is near 0.045, which means that 4.5% of the material volume is composed of voids, occurring for *a*_0_ = 21.5 mm and a plastic strain of about 110%.

The increase in initial crack length tends to increase the porosity growth rate, which means that for the same plastic strain there is more porosity. The higher initial crack lengths induce higher Δ*K* values, which result in higher porosity levels at the instant of node release. The initial values of porosity also depend on initial crack length. Note that the numerical model works with a discrete propagation scheme: at the critical plastic strain the node containing the crack tip is released. Thus, when propagation occurs, the crack tip advances, moving away from the highly strained zone. As a TPS approach is followed when propagation occurs, the plastic strain and porosity occurring at the node which was previously ahead of the crack tip, and now contains it, are assumed. However, this reset drives the referred entities to different values. For *a*_0_ = 21.5 mm, both plastic strain and porosity are higher than for the remaining initial crack lengths. On its way, for *a*_0_ = 11.5 mm, only porosity is set to a higher level than for the lower initial crack length. This occurs because higher stress intensity factors result in higher plastically affected zones, and higher strains. This way, when the crack advances it reaches differently affected zones, explaining the obtained values of porosity and plastic strain. The successive load cycles cause the porosity to gradually grow. Therefore, the premise that the build-up of plastic strain causes an accumulation of plastic damage is verified.

Another interesting detail perceptible in Figure 11 is the fact that porosity shows an oscillating behaviour. This is more perceptible for *a*_0_ = 21.5 mm, due to the higher oscillations registered, but it also occurs for the remaining values of *a*_0_. During the unloading phase of each loading cycle the stress verified at the crack tip is of compressive nature. This stress causes the micro-voids on the material to partially close and consequently, the porosity is reduced. Nevertheless, the micro-cavities do not disappear since the damage is irreversible.

### 3.6. Stress Triaxiality

The differences in the evolution of the porosity with the plastic strain can be explained by the stress triaxiality at the crack tip. Indeed, using the GTN damage model, the void fraction evolution is significantly affected by the stress triaxiality [28]. The present model only considers the growth of micro voids, this process being highly influenced by the stress triaxiality [44]. Figure 12a presents the evolution of the stress triaxiality at the crack tip during the propagation shown in Figure 4, comparing three different crack lengths; Figure 12b presents analogous results but for the porosity. Both results (stress triaxiality and plastic strain) were predicted at the maximum load instant. The horizontal axis denotes the progress up to propagation, making it possible to compare propagations with different lengths of time. Higher Δ*K* generate higher porosity levels; therefore, the relative position of the curves is not a surprise. However, note that stress triaxiality is initially very high for *a*_0_ = 5 mm, generating a fast increase in porosity, as can be seen in Figure 12b. Stress triaxiality then stabilises coinciding with the saturation of porosity. For *a*_0_ = 11.5 mm the stress triaxiality at the beginning is lower, corresponding to a less abrupt increase in porosity. Additionally, stress triaxiality suffers a much less significant drop, which can explain the inexistence of saturation for this *a*_0_. For the higher initial crack length, the stress triaxiality is relatively high and increases during the propagation, matching the higher slope attained for porosity. As a result, it may explain the trends followed by porosity.

## 4. Discussion

Past simulations of *da/dN* were based solely on the plastic deformation as a driving force, which is independent of mean stress. The inclusion of the nucleation and growth of micro-voids is a step towards a better understanding of FCG. In fact, the existence of intrinsic defects may be expected, resulting from technological processes such as casting or additive manufacturing. Additionally, voids nucleate by debonding of the second phase particles.

Figure 13 compares experimental results of *da/dN* with numerical predictions with the ones obtained with and without the GTN model. Without the inclusion of the growth of micro-voids, the numerical model underestimates the slope of the *da/dN-*Δ*K* curve in log-log scales, as shown by the dotted line. With GTN there is an anti-clockwise rotation of the curve approximating it to the experimental results, as highlighted by the full line. Note that the Paris–Erdogan law *m* parameter is 3.62, which is still higher than the ones obtained with GTN (*m =* 3.36) and without GTN (*m =* 2.62). However, the model with GTN provides a slope closer to the experimental results.

The presence of voids in regime II of FCG is, however, a subject of discussion. In some cases the voids are clearly visible [7]. On the other hand, in many other situations the voids are not clearly visible. However, this mechanism certainly does not work in an on/off mode. In other words, in many situations there will certainly be voids which are not visible. There are technological processes which favour the presence of defects, namely additive manufacturing, casting and welding. Additionally, second-phase particles favour the formation of pores [45]. Finally, the anti-clockwise rotation of the *da/dN-*Δ*K* curve may be seen as indirect proof of the presence of micro-voids.

The CTOD analysis, presented in Figure 7, also allowed us to assess the validity of the small scale yielding (SSY) condition, which certifies the utilization of the Δ*K* parameter. SSY conditions were shown to dominate when the elastic component of CTOD (Δ*δ*_e_) was larger than 75% of the total CTOD (Δ*δ*_t_), measured at a distance of 8 μm from the crack tip [46], i.e., at the node behind the crack tip, as it is the case in the conducted analysis. Thus, if Δ*δ*_e_/Δ*δ*_t_ > 75% the SSY condition is verified. For *a*_0_ = 11.5 mm the aforementioned ratio is 85.6% in the case of the model with GTN and 81.1% in the version without GTN. Analogously, for *a*_0_ = 21.5 mm, with GTN the ratio is 58.4% and without GTN 59.7%. In this manner, SSY ceases to dominate for the higher crack length; therefore, the Δ*K* may be questioned. Note, however, that the FCG analysis is based on the crack tip plastic deformation and Δ*K* is used just to represent the findings. Moreover, the eventual limited validity of Δ*K* is not expected to affect the global trends, namely the comparison between GTN and non-GTN models.

## 5. Conclusions

Fatigue crack growth is predicted here assuming that cyclic plastic deformation at the crack tip is the fatigue crack growth driving force. The growth of micro-voids was included in the analysis, providing a better modelling of crack tip damage. The main conclusions are:
The inclusion of micro-voids in the model based on cumulative plastic strain produced an unexpected decrease in *da/dN* for low values of Δ*K*. On the other hand, at relatively high values of Δ*K*, the GTN model increased the FCG rate.The inclusion of porosity in the analysis increases the plastic deformation level at the crack tip as well as the size of the plastic zones ahead of the crack tip.This higher plastic deformation results in higher plastic wakes at the crack flanks, decreasing the crack opening level.At low values of Δ*K*, the inclusion of micro-voids increased plasticity induced crack closure (PICC), promoting the reduction in *da/dN*. At high values of Δ*K*, there is no PICC even with GTN. Therefore, the variations of *da/dN* are linked with changes of PICC. Disabling the contact of crack flanks results in an increase in *da/dN* with GTN, for all values of Δ*K* studied.There is a global trend for the increase in porosity with plastic strain. However, an oscillatory behaviour is observed in each load cycle, since the stress at the crack tip is of compressive nature during the unloading phase. This induces a partial close of the micro-voids on the material. The increase in crack length and, therefore, of Δ*K*, also increases the porosity level.Regardless, the variation of porosity with plastic strain is relatively complex. This complexity was explained by the strong link found between stress triaxiality and porosity level.

## Figures and Tables

**Figure 1 materials-14-04303-f001:**
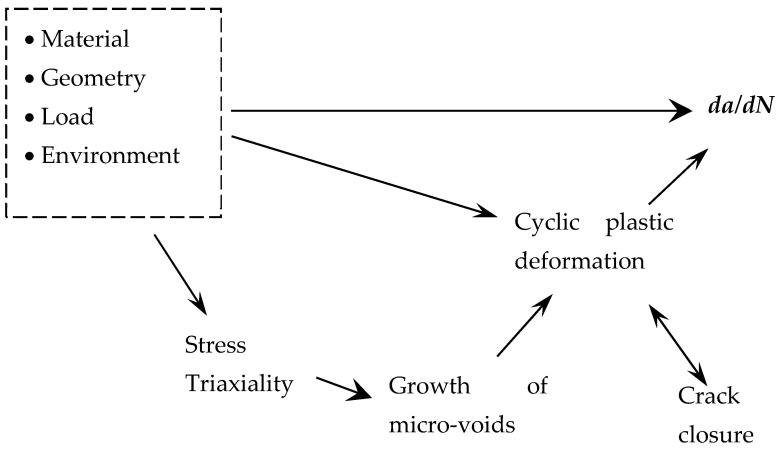
Schematic illustration of mechanisms and parameters involved in FCG modelling.

**Figure 2 materials-14-04303-f002:**
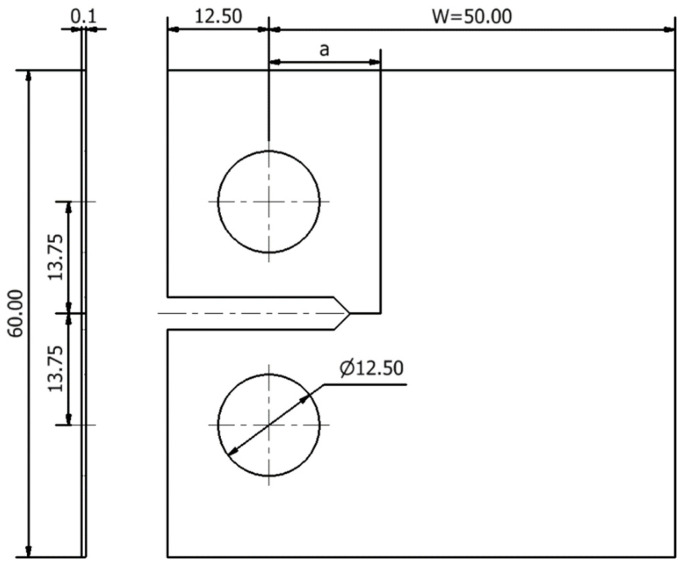
Compact tension specimen modelled for AA2024-T351, with dimensions in mm.

**Figure 3 materials-14-04303-f003:**
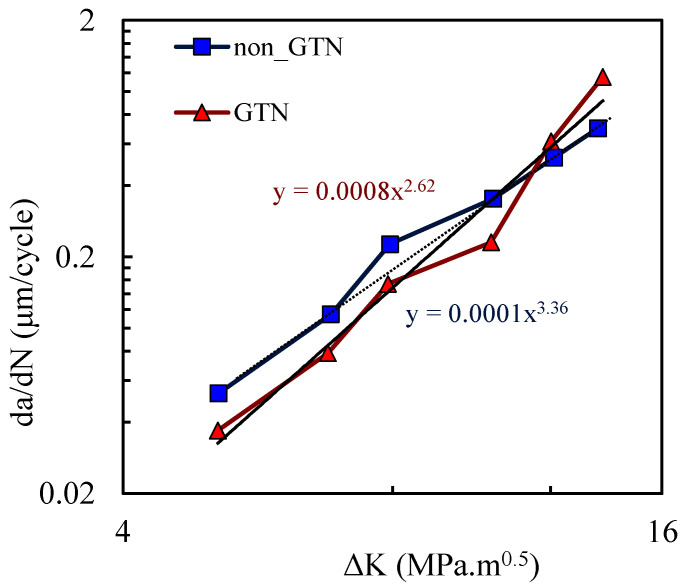
*da/dN*-*ΔK* curves in log-log scale (plane strain; *R* = 0.1; *f*_0_ = 0.01; *q*_1_ = 1.5; *q*_2_
*=* 1 and *q*_3_
*=* 2.25, nucleation and coalescence are disabled). The Paris–Erdogan law parameters are shown on the equations related to the trendlines.

**Figure 4 materials-14-04303-f004:**
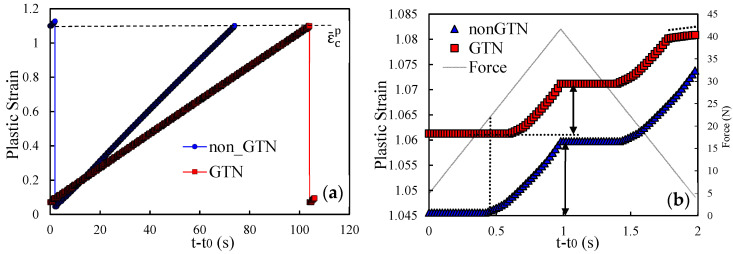
Comparison of the plastic strain evolution with and without GTN for *a*_0_ = 11.5 mm. (**a**) Time period between the 25th and the 26th node releases. (**b**) A single load cycle, immediately before the 26th propagation.

**Figure 5 materials-14-04303-f005:**
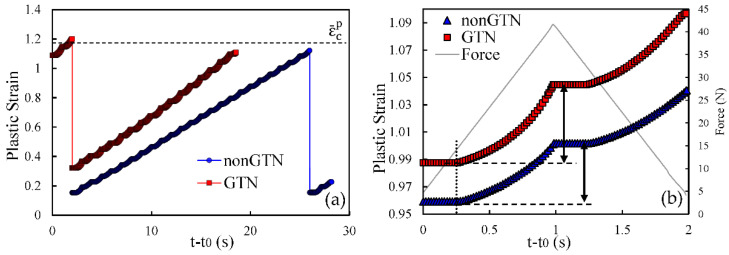
Comparison of the plastic strain evolution with and without GTN for *a*_0_ = 21.5 mm. (**a**) Time period between the 36th and the 37th node releases. (**b**) A single load cycle, immediately before the 37th propagation.

**Figure 6 materials-14-04303-f006:**
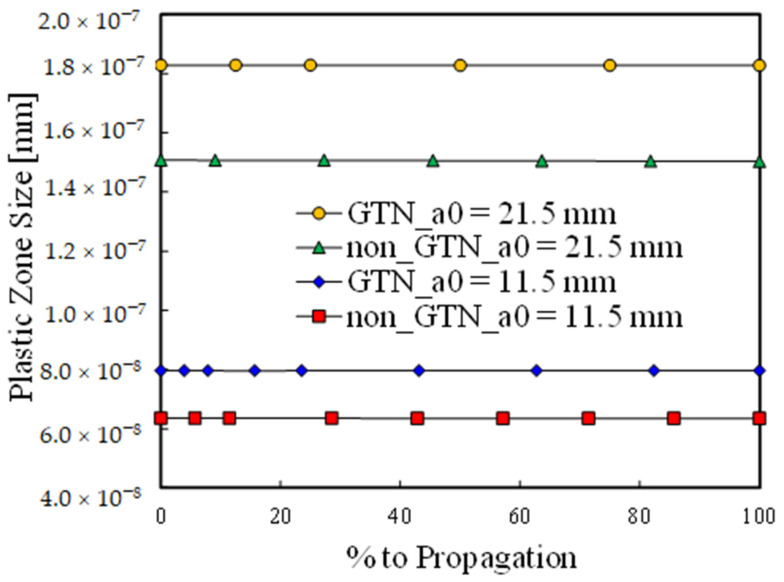
Size of the plastic zone at the crack tip evaluated for *a*_0_ = 11.5 mm and *a*_0_ = 21.5 mm considering both models—with and without GTN.

**Figure 7 materials-14-04303-f007:**
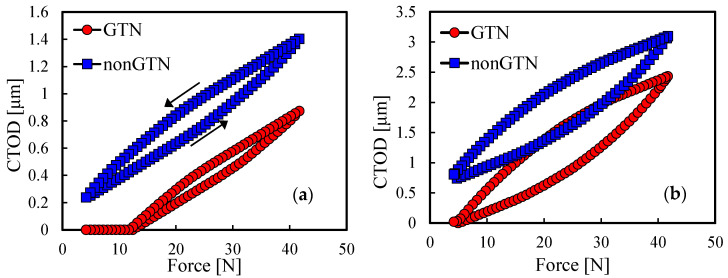
Comparison of CTOD predicted with and without GTN for: (**a**) *a*_0_ = 11.5 mm, at the same load cycle of Figure 4b; (**b**) *a*_0_ = 21.5 mm, at the same load cycles of Figure 5b (plane strain).

**Figure 8 materials-14-04303-f008:**
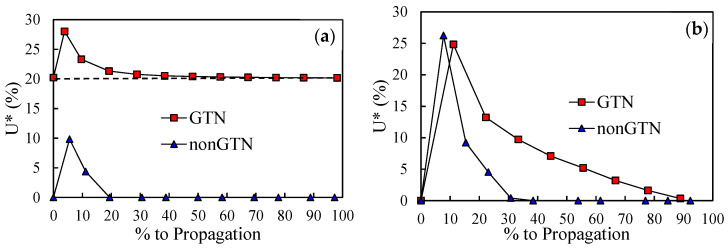
Crack closure level with and without GTN. (**a**) *a*_0_ = 11.5 mm, between the 25th and 26th crack propagations. (**b**) *a*_0_ = 21.5 mm, between the 36th and 37th crack propagations. The results are presented in percentage up to propagation, which is an analogous scale to the pseudo time.

**Figure 9 materials-14-04303-f009:**
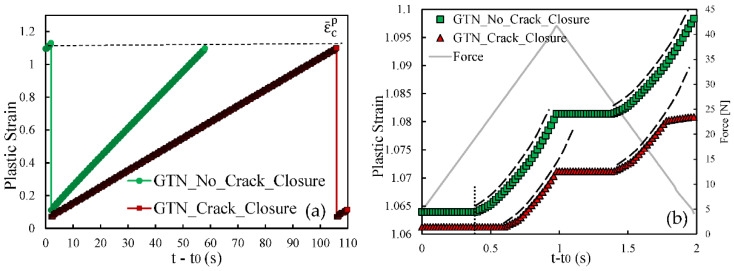
Effect of crack closure on plastic strain evolution, for *a*_0_ = 11.5 mm. (**a**) Period between the 25th and the 26th crack propagations. (**b**) A single load cycle, before the 26th crack propagation.

**Figure 10 materials-14-04303-f010:**
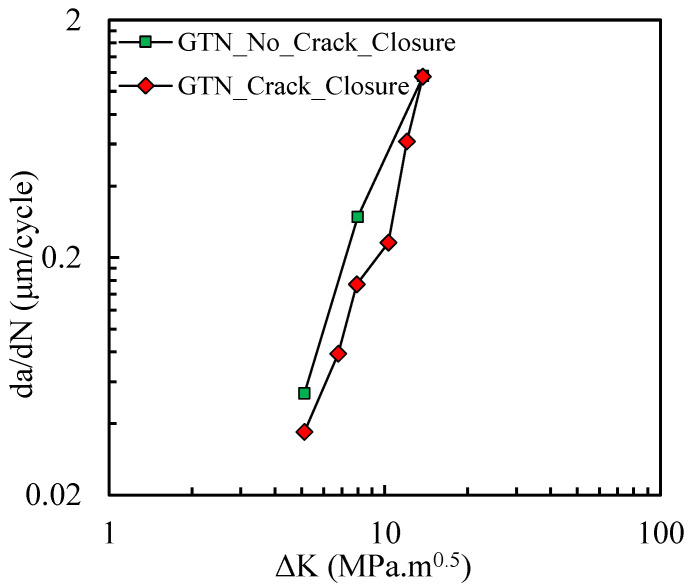
Effect of crack closure on *da/dN* values (model with GTN).

**Figure 11 materials-14-04303-f011:**
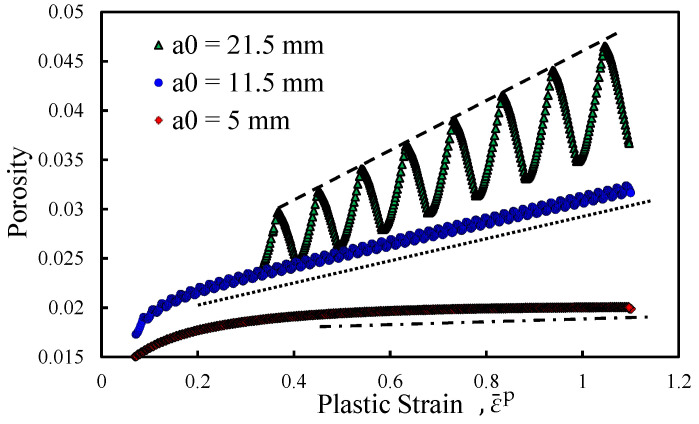
Porosity evolution with plastic strain growth for different initial crack lengths (*a*_0_) in natural scales. Crack closure is enabled.

**Figure 12 materials-14-04303-f012:**
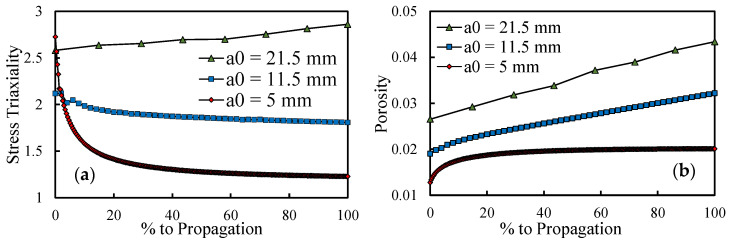
(**a**) Stress triaxiality throughout the entire propagation studied in Figure 4. (**b**) Porosity evolution for the same propagation.

**Figure 13 materials-14-04303-f013:**
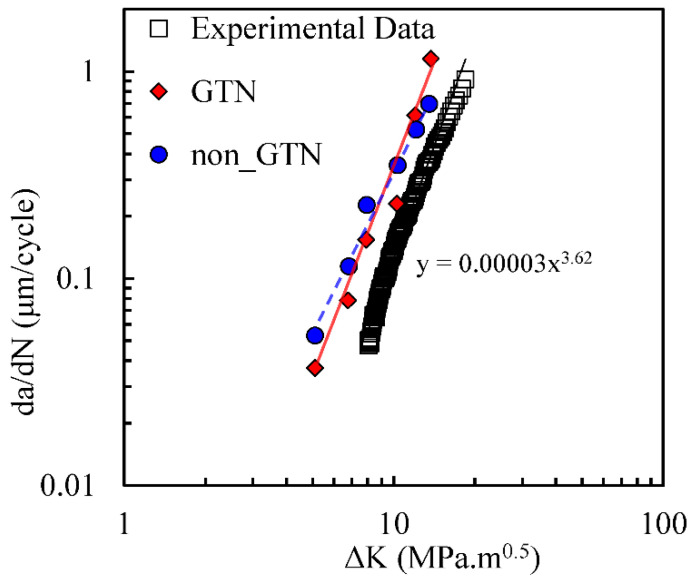
*da/dN*-Δ*K* curves in log-log scale (plane strain; *F*_min_ = 4.17 N; *F*_max_ = 41.7 N; *R* = 0.1). The Paris–Erdogan law parameters are shown on the equation related to the trend-line added to the experimental results.

**Table 1 materials-14-04303-t001:** Elastic properties and Swift and Armstrong–Frederick laws’ parameters obtained for the 2024-T351 aluminium alloy.

Material	*E* (GPa)	*ν* (-)	*Y*_0_ (MPa)	*k* (MPa)	*n* (-)	*X*_Sat_ (MPa)	*C_X_* (-)
AA2024-T351	72.26	0.29	288.96	389.00	0.056	111.84	138.80

**Table 2 materials-14-04303-t002:** GTN parameters adopted in the numerical model. Note that nucleation was neglected.

Material	*f* _0_	*q* _1_	*q* _2_	*q* _3_
AA2024-T351	0.01	1.5	1	2.25

## Data Availability

Data sharing not applicable.

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
