# Peer review of "FCG Modelling Considering the Combined Effects of Cyclic Plastic Deformation and Growth of Micro-Voids"

_materials, 2021, doi:10.3390/ma14154303_

Round 1

Reviewer 1 Report

This study discusses the fatigue crack growth model with the Gurson-Tvergaard-Needleman (GTN) damage model. Please consider the following points.

Major Comments

  1. Please specify what "damage" accumulate at the crack tip used in this manuscript. Dislocations, vacancies, voids, plastic strain, and plastic deformation are possible. Since it refers to the GTN model, the reviewer assumes it is void, but it is not consistent with "assuming that cyclic plastic deformation at the crack tip is the main damage mechanism."
  2. In the fatigue crack growth fracture surface, there is evidence of the fatigue crack mechanism in which the fatigue crack grows one time in one load cycle by plastic deformation in the form of striation. There is no evidence to suggest that microvoid growth plays a role in the fatigue crack growth mechanism. Therefore, if there is any evidence that the growth of microvoids is affected, please discuss it. If you think that microvoids exist but are too small to be observed, discuss them quantitatively. However, reviewers do not believe that fatigue crack growth is the mechanism described above for all materials and all crack growth rate domains. The presence of conditions affected by microvoid growth cannot be denied. Then, please discuss the condition in which the growth of the microvoid affects.
  3. In order to discuss the fatigue crack growth using the stress intensity factor, it is necessary to discuss whether the small-scale yield condition is satisfied or not. Please discuss it.
  4. For FEM analysis, is the analysis model two-dimensional or three-dimensional? From Fig.2, it is 0.1 mm thickness; therefore, the reviewer guesses it's three-dimensional. However, in this case, "only plane strain conditions were observed" is unclear. How was plane strain condition established in three dimensions?
  5. Also, for FEM analysis, it seems that the stress intensity factor cannot be accurately expressed with the element size of 8 um in the crack extension region for this crack length used in this study. Please provide evidence "which allows to accurately evaluate the strong gradients of stresses and strains at this zone."

Minor Comments

  1. Please reconfirm the numbers in the text and captions.
  2. If a sentence follows a formula, the line is not indented.
  3. Which plastic strain is shown on the horizontal axis in Figure 11? Subscript should be indicated in subscript. Please indicate the unit.
  4. It is not clear what the red and blue lines in Figure 13 (Figure 6 in the caption) represent. Also, for comparison with other results, the scale on the horizontal axis should be 1, 10, 100, not 1, 4, 16, 64.

Reviewer 2 Report

Review of the manuscript: FCG modelling considering the combined effects of cyclic plastic deformation and growth of micro-voids

In the present manuscript, the Authors have investigated a numerical model that uses the cyclic plastic strain at the crack tip to predict da/dN coupled with the Gurson-Tvergaard-Needleman (GTN) damage model. The GTN model has been adopted to account for the material degradation due to the growth of micro-voids process. Crack closure has also been studied. Predictions have been compared with experimental results.

The topic treated in the manuscript is interesting and the paper is well written and structured, however minor revisions are necessary to improve the manuscript and to publish it.

Minor issues:

  • A list of symbols and abbreviations should be added to help the reader.
  • Authors should discuss if the considered specimen geometry is in agreement with either ASTM or ISO standards. Moreover, what is the notch opening angle and the notch tip radius? The notch tip radius has been accounted for in the FE modelling, especially in the case of short crack length propagated at the notch tip?
  • Dealing with FE type, the integration option should be reported in the text: is it a full or reduced or other formulation?
  • Unit of measurements should be added to Eq. (7) when adding the value ‘8’ which is not a pure number.
  • Figures should be re-order. Please check all the captions.
  • The size of the text inside the figures should be increased to improve readability.
  • Other minor issues are reported in the marked-up manuscript.

Reviewer 3 Report

In this paper, the authors present the results of systematic numerical studies of the fatigue crack growth process, using the cyclic plastic deformation model at the tip of the crack to predict the da / dN coupled with the Gurson-Tvergaard-Needleman (GTN) failure. The process of crack propagation occurs by releasing nodes when the cumulative plastic deformation reaches a critical value.

The introduction and experimental parts are well written, and the authors detailed the influence of cyclic plastic deformation on deformation propagation. The model takes into account the existence and growth of microvessels, which makes the results of numerical studies closer to the results of experimental tests.

However, please clarify:

- where the model takes into account the influence of the temperature of the deformation process,

- does the mesh adopted in modeling reflects the actual structure of the tested material?

- It is worth explaining whether the discussed model will also be applicable to the cyclic deformation of other Al alloys.

Round 2

Reviewer 1 Report

The final sentence of the discussion: Please consider the number of significant figures. 85.64%, 81.06%, 58.39%,  and 59.65% should be 85.6%, 81.1%, 58.4% and 59.7%, respectively.
